Reduction of Cr(VI) by Bacillus toyonensis LBA36 and its effect on radish seedlings under Cr(VI) stress

Tan Aobo 1
Wang Hui 1 wanghui79@haust.edu.cn
Zhang Hehe 1
Zhang Longfei 1
Yao Hanyue 1
Chen Zhi 2
1 Department of Environmental Engineering, College of Chemistry and Chemical Engineering, Henan University of Science and Technology , Luoyang, Henan , China
2 Department of Civil and Environmental Engineering, Concordia University , Montreal , Canada
Wang Liang
Electronic publication date: 2024 Sep 24
Publication date: 2024
Volume: 12
Electronic Location ID: e18001
Received 2024 Mar 28; Accepted 2024 Aug 7
Copyright: © 2024 Tan et al.
Copyright year: 2024
Copyright holder: Tan et al.
License: This is an open access article distributed under the terms of the Creative Commons Attribution License, which permits unrestricted use, distribution, reproduction and adaptation in any medium and for any purpose provided that it is properly attributed. For attribution, the original author(s), title, publication source (PeerJ) and either DOI or URL of the article must be cited.
License URL: https://creativecommons.org/licenses/by/4.0/

Keywords: Cr-tolerant microorganisms, Bacillus toyonensis, Cr reduction, Hydroponics, Radish seedlings, Microbial-phytoremediation

Funding: National Natural Science Foundation of China 51975186 Henan Provincial Higher Education Institutions 2018GGJS047 Henan Provincial Science and Technology 192102110050 This work was supported by the National Natural Science Foundation of China (51975186), the Henan Provincial Higher Education Institutions Young Backbone Teacher Training Program (No. 2018GGJS047) and the Henan Provincial Science and Technology Tackling Project for the Development of Microbial In situ Remediation of Heavy Metal Pollution in Agricultural Soil in Mining Areas (192102110050). The funders had no role in study design, data collection and analysis, decision to publish, or preparation of the manuscript.

==============================
Chromium, being among the most toxic heavy metals, continues to demand immediate attention in the remediation of Cr-contaminated environments. In this study, a strain of LBA36 (Bacillus toyonensis) was isolated from heavy metal contaminated soil in Luanchuan County, Luoyang City, China. The reduction and adsorption rates of LBA36 in 30 mg·L−1 Cr-containing medium were 97.95% and 8.8%, respectively. The reduction mechanism was confirmed by Fourier-transform infrared spectroscopy, and X-ray photoelectron spectroscopy (XPS). Cr(VI) reduction by this strain predominantly occurred outside the cell, with hydroxyl, amide, carboxyl, C-N group, carbonyl, and sulfur carbonyl as the main reaction sites. XPS analysis revealed the presence of Cr2p1/2 and Cr2p3/2. Furthermore, the hydroponic experiment showed that the fresh weight and plant height of radish seedlings increased by 87.87% and 37.07%, respectively, after inoculation with LBA36 strain under 7 mg·L−1 Cr(VI) stress. The levels of chlorophyll, total protein, malondialdehyde, superoxide dismutase and catalase were also affected to different degrees. In conclusion, this study demonstrated the potential of microbial and phytoremediation in the treatment of heavy metal toxicity, and laid the foundation for the development of effective bioremediation methods for Cr(VI) pollution.

Introduction

Chromium (Cr) is a heavy metal, and environmental Cr pollution from anthropogenic activities, including industrial activities such as mining, metallurgy, tanning, and textile production, has become increasingly severe. Waste generated via industrial production is directly discharged into water bodies or soil without treatment, contributing to soil and groundwater contamination and harming the ecosystem (Chen et al., 2022). Cr has various oxidation states (0–VI); however, only two valence states, trivalent and hexavalent, are stable (e.g., chromite (Cr(III)) and chromate (Cr(VI))), and the differences in oxidation states lead to changes in the properties and toxicity of the metal (Ayele, Godeto & Zhang, 2021). Cr(VI) is designated as a priority pollutant owing to its carcinogenicity and mutagenicity (Tirry et al., 2018). Cr(VI) induces toxicity in several ways; it can reduce the activity or efficiency of the immune system and induce changes in cellular structure, especially in the lipoprotein region of the cell membrane. Furthermore, Cr(VI) can enter the human body through inhalation, ingestion, and dermal contact (Sharma et al., 2022). In contrast, Cr(III) derivatives are less hazardous and mutagenic due to their insolubility in water at a neutral pH. Therefore, the reduction of Cr(VI) to Cr(III) is a crucial Cr(VI) remediation strategy.

Numerous methods exist to convert Cr(VI) to Cr(III), such as chemical transformation, redox treatment, coagulation, photocatalytic and electrochemical reduction, bioremediation, and nanotechnological interventions (Imran et al., 2021; Yasmin et al., 2021). The redox method is among the commonly used methods in chemical transformation, where Cr(VI) is reduced to the less toxic form of Cr(III), followed by rapid precipitation as Cr hydroxide or the solid compound FexCr1_x(OH)3; and sulfur-or iron-based salts can be used as chemical reducing agents during this process (Prasad et al., 2021). The chemical conversion method has disadvantage in that it generates a significant amount of toxic sludge, thereby causing secondary pollution (Lian et al., 2019). In contrast, electrokinetic remediation, is a relatively effective and environmentally friendly technique, in which low-voltage direct current is passed into the soil, prompting the migration of the toxic Cr(VI) towards the electrodes via electromigration. The Pd/Fe3O4 nanocatalyst used by Huang et al. (2017) was highly efficient in eliminating Cr(VI) via the electro-Fenton method. Mu’azu et al. (2016) achieved 89.64% Cr remediation in bentonite by applying an improved electrokinetic remediation technique called pulsed electrokinetic remediation (PEKR). Nanoparticles have been used as adsorbent materials owing to their high specific surface area (availability of active sites), and high reactivity and reduction capacity (Souza, Pomarolli & da Veiga, 2020). One gram of Fe nanoparticles can reduce 69.3–72.7 mg of Cr(VI) in chromite ore, and up to 99% of Cr-contaminated soil can be remediated with 5 g·L−1 of Fe nanoparticles (Cao et al., 2016; Singh, Misra & Singh, 2011). While both electric remediation and nanotechnology have disadvantages such as high energy consumption, complex processes, high cost, and secondary pollution production during the remediation process.

Considering the drawbacks of physicochemical remediation of Cr contamination, bioremediation is considered a convenient and cost-effective method that does not cause secondary contamination. Various microorganisms have been utilized in Cr remediation; these include a strain of Pseudomonas aeruginosa Cr13, which exhibited a tolerance of 250 mg·L−1 and 94.26% removal of heavy metal Cr from Cr-contaminated soil (Gong et al., 2020). Prabhakaran et al. (2019) remediated Cr(VI) using the biosorption and bioreduction mechanisms of Sphingopyxis macrogoltabida Suk2 strain. In addition, other strains, including Escherichia sp. (Mohamed et al., 2020), Caldicellulosiruptor saccharolyticus (Bai et al., 2018), Halomonas sp. DK4 (Kalola & Desai, 2019), and P. aeruginosa (Noureddine et al., 2016) have exhibited very high Cr bioremediation potential. Phytoremediation techniques have also been applied for soil Cr remediation (Souza, Pomarolli & da Veiga, 2020). Researchers have used hyperaccumulating plants for phytoextraction of Cr from soil (Levizou, Zanni & Antoniadis, 2018). Origanum vulgare demonstrated an accumulation capacity of 4,300 mg·kg−1 of Cr, Cichorium spinosum could accumulate 330 mg·kg−1 (Antoniadis et al., 2017), and Vigna mungo accumulated 74.22 ± 2.04 mg·kg−1 (Saravanan et al., 2019). However, although these plants can act as bioconcentrators, the presence of Cr imposes stress on them, affecting their growth and reducing their remediation efficacy.

The use of microorganisms in conjunction with plants can combine the advantages of both approaches, with microbial action reducing Cr stress in plants and enhancing phytoremediation processes. For example, Tirry et al. (2018) isolated 27 strains of Cr-tolerant bacteria and conducted greenhouse pot experiments to demonstrate that inoculating plants with these strains enhanced the growth of alfalfa plants and increased the uptake of heavy metals by these plants grown. Ontañon et al. (2014) conducted a study to increase the heavy metal uptake in oilseed rape by utilizing antoe asp. FC 1, a phenol-and Cr-tolerant strain. The study revealed that antoe asp. FC 1 exhibited restorative properties and can assist in phytoremediation when used in conjunction with oilseed rape. Maqbool et al. (2015) screened a strain of Brucella k12 that imparts high Cr resistance and Cr-lowering capacity to okra. They demonstrated that inoculating okra with this strain can reduce the toxic effect of Cr on okra, thereby promoting growth and yield of okra. Cruciferous plants have been reported to absorb and transfer high levels of chromium to edible parts (Cervantes et al., 2001; Singh et al., 2013). Radish belongs to the family of cruciferous plants which can be easily grown in various climates and is considered as a model crop (Buttery & Buzzell, 1977). And there are different studies correlating radish with heavy metals. Zaman & Zereen (1998) found that radish showed different levels of sensitivity to different concentrations of Cd or Pb in soil and hence these plants can be used as bioindicators of Cd and Pb in metal contaminated soil. Yadav, Shukla & Sharma (2009) investigated the effect of different concentrations of nickel on radish germination, growth, biomass, and levels of several enzymes. It was found that nickel treatment resulted in a significant decrease in germination, growth and biomass. Therefore, radish was used as an experimental subject in the present study.

Although several strains have been reported with the ability to reduce Cr(VI) levels, there is a need to identify new microbial resources to improve crop growth and yield in Cr-contaminated soils and to reduce Cr(VI) to Cr(III). Therefore, the aim of this study was to screen Cr(VI)-reducing bacteria in Cr-contaminated soil, and investigate their growth pattern under Cr-contaminated conditions and capacity to remove Cr. Furthermore, we investigated the effect of these strains on the growth of radish seedlings under Cr stress.

Materials and Methods

Soil sample

The soil used for microbial isolation was obtained from the molybdenum and lead mining area in Luanchuan County, Henan Province (111°54′E, 33°82′N) (Fig. 1). The soil texture was sandy loam with a pH of 6.5, and the heavy metal content of the soil is shown in Table 1.

Figure 1 Geographic location of the sampling area.

The data set is provided by Geographic remote sensing ecological network platform (http://www.gisrs.cn/).

Table 1 Concentrations of heavy metals in the soil.

Heavy metals	Content (mg·kg−1)	Heavy metals	Content (mg·kg−1)	
Chromium	68.9	Copper	52.3	
Cadmium	6.66	Lead	1,018	
Nickel	24.7	Iron	39,943	
Mercury	0.043	Zinc	1,340	
Arsenic	16.8			

Isolation, screening and identification of Cr(VI)-resistant bacterial strain

One gram of soil was used to prepare a 100 mL soil suspension, followed by magnetic stirring and settling. Five milliliters of the supernatant were collected and added to Luria-Bertani (LB) broth containing 30 mg·L−1 Cr(VI) (K2Cr2O7), and incubated in shaking conditions at 30 °C and 150 r·min−1 to obtain a bacterial suspension. Bacterial suspension (1 mL) was spread onto LB agar containing 30 mg·L−1 Cr(VI) (30 °C, 24 h). Individual colonies that appeared were selected for cultivation. This process was repeated multiple times until the Cr(VI)-resistant bacterial strain was identified.

The DNA of the strain was extracted using the boiled template method, while the amplification of the 16S rDNA gene was conducted using polymerase chain reaction (PCR) with primers 27F (5′-AGAGTTTGATCMTGGCTCAG-3′) and 1492R (5′-ACGGCTACCTTGTTACG A-3′) (Zhang et al., 2022). The temperature cycling program was configured to preheat at 94 °C for 45 s, annealing at 56 °C for 45 s, and extension at 72 °C for 90 s. This cycling process was repeated 30 times, followed by a final extension at 72 °C for 10 min (Ji et al., 2018). The PCR amplification system is shown in Table 2. The PCR products were sequenced, and the results were compared with the data from the National Center for Biotechnology Information database and EZBioCloud database to construct a phylogenetic tree (Loong et al., 2016).

Table 2 PCR amplification system.

Reaction system	Volume (µL)	Reaction system	Volume (µL)	
ddH2O	9.5	2 × Taq DNA polymerase	12.5	
Primer 27F	1	DNA template	1	
Primer 1492R	1	Overall system	25	

Minimum inhibitory concentration, chromium reduction, and adsorption experiments

The LBA36 strain was inoculated in LB medium at concentrations of 0, 30, 60, 90, 120, and 150 mg·L−1 and incubated for 88 h. The OD600nm value of the culture solution was measured at regular intervals to determine the minimum inhibitory Cr concentration (MIC) of the LBA36 strain. Three replications were set up for each set of experiments.

LBA36 strain was cultured in medium containing Cr(VI) 30, 60, and 90 mg·L−1 to investigate the removal of heavy metals by LBA36 strain at different Cr(VI) concentrations (the minimum inhibitory concentration experiments showed that the LBA36 strain had different growth characteristics under Cr(VI) stress of 30, 60, and 90 mg·L−1). Cr(VI)-containing medium without strain inoculation was used as a negative control. Three parallel groups were set up for each concentration. Samples were collected, incubated for 32 h (8,000 r·min−1, 10 min) and then centrifuged, and 1 mL of supernatant was extracted for digestion. The LBA36 strain was cultured in medium containing Cr(VI) 30 mg·L−1 to explore the effect of time on Cr(VI) removal by LBA36 strain. Cr(VI)-containing medium without strain inoculation was used as a negative control. The incubation period was 56 h and samples were taken every 8 h. Three replications were set up for each set of experiments.

Total chromium quantification can only be performed using a digested supernatant solution (Dong et al., 2022). The supernatant underwent digestion using the electrothermal plate method (Dong et al., 2022). One milliliter of the collected supernatant was transferred into a polytetrafluoroethylene crucible, to which 5 mL of HCl was added. Low-temperature (110 °C) digestion was conducted on an electric heating plate until the volume was reduced to 1–2 mL. Subsequently, 5 mL of HNO3 was introduced for high-temperature (160 °C) digestion, and once the volume reached 1–2 mL, the temperature was lowered for acid evaporation until complete sample digestion was achieved. The digested sample was then dissolved in a 50 mL volumetric flask, supplemented with 1 mL of HNO3 to ensure a 2% acidity level, and finally brought to volume with ultra-pure water (50 mL). Thus, total chromium content was measured in the digested solution using an inductively coupled plasma spectrometer (ICPS) (Astatkie, Ambelu & Beyene, 2021). The Cr(VI) content of the samples was determined spectrophotometrically using dibenzoyl dihydrazide (Ribas et al., 2022). The reduction rate of Cr(VI) by the strain was calculated with the following formula:

(1) q1=(q2−q3)q2

where q1 is the reduction efficiency of Cr(VI) by the strain; q2 is the supernatant Cr(VI) content of the control group; and q3 is the supernatant Cr(VI) content of the experimental group.

The adsorption rate of total chromium by the strain was calculated with the following equation:

(2) Q1=(Q2−Q3)Q2

where Q1 is the adsorption efficiency; Q2 is the total chromium content in the supernatant of the control group; and Q3 is the total chromium content in the supernatant of the experimental group.

Assessment of the reducing properties of various cellular components

Extracellular and intracellular materials and bacterial debris of the bacteria were extracted, and they were subsequently divided into resting and permeable cells based on treatment methods (Han et al., 2020). The isolated bacterial strain was cultured for 32 h. The culture fluid was then aspirated and centrifuged (6,000 r·min−1, 4 °C, 20 min). The resulting supernatant was collected as the extracellular material. The untreated supernatant sample was termed the “resting sample”. The supernatant to which 1% toluene and 0.2% Triton-x100 was added was called the “permeabilized sample” (Megharaj, Avudainayagam & Naidu, 2003). The intracellular and cellular debris samples were generated as follows: the extracellular supernatant sample was broken by ultrasonication in an ice bath (20 s on, 30 s off, 200 W, 20 min duration), and the broken sample was centrifuged at 4 °C, 12,000 r·min−1 for 30 min, and the supernatant was collected after centrifugation as the intracellular material (intracellular extract) along with cellular debris. The cell debris was suspended in 10 mL of 25 mM Tris-HCl buffer (pH = 7.0). Tris-HCl buffer (10 mL, 25 mMol, and pH = 7.0) was used as a blank instead of cell fractions. One milliliter each of the above-mentioned samples was added separately to culture media containing 10 mg·L−1 Cr(VI), which was then inoculated with LBA36 strain and incubated for 6 h. The sample was then centrifuged and the Cr(VI) remaining in the supernatant was used to determine the reducing properties of the different cellular components.

Chemical analysis and characterization

The morphology of the bacterial strain was assessed using scanning electron microscopy (SEM). The bacterial solution was centrifuged (7,000 r·min−1, 10 min), the bacterial body was collected and dehydrated using an ethanol gradient (Wang et al., 2019). The ethanol was allowed to evaporate, and the bacterial powder was coated on the sample stage and sprayed with gold. Bacterial cell images were observed using a scanning electron microscope.

LBA36 cells were cultured in media containing Cr(VI) 0 mg·L−1, 30 mg·L−1, and 60 mg·L−1 for 32 h. The culture medium was taken and centrifuged to collect the cells, which were dried in a vacuum freeze dryer (An et al., 2022; Zeng et al., 2020). After drying, Fourier-transform infrared spectroscopy (FTIR), and XPS were used to investigate the changes in surface functional groups of the LBA36 strain and Cr(VI) valence before and after Cr(VI) stress.

Radish seedling hydroponic experiment

The radish seedling hydroponic experiment was performed to investigate the effects of LBA36 on the growth and development of radish seeds as well as their physiology and biochemistry in the presence of Cr(VI) ions. The radish used in the experiment was sweet and crispy white radish seeds (cultivated seeds) of Botou Yonghong Seed Co. The general-purpose hydroponic nutrient solution used in the experiment consisted of Ca(NO3)2, KNO3, NH4H2PO4, MgSO4, Na2B4O7, MnSO4, (NH4)2MoO4, CuSO4 and ZnSO4.

In vivo experiments: Sterilized radish seeds were soaked separately in a culture solution containing LBA36 and sterile water for 2 h. Sterile filter papers were soaked in Cr(VI) solution at graded concentrations (0, 30, 60, 90, 100, 200, and 400 mg·L−1). The filter papers corresponding to different Cr(VI) concentrations were then placed in petri dishes, and 50 radish seeds were placed in each petri dish. To minimize variability in other conditions during incubation, petri dishes were incubated in the dark at 30 °C for 7 days. Sterile distilled water was added every 24 h. After the seeds germinated, the germination percentage was calculated, and the seedling diameter length, root length, seedling fresh weight, and dry weight were measured. The experiment was repeated thrice for each Cr(VI) concentration level.

In vitro experiment: White radish seeds were disinfected with a 10% sodium hypochlorite solution for 10 min, and then rinsed with sterile distilled water 3–4 times. Subsequently, the disinfected white radish seeds were soaked in sterile distilled water for 6 h. The soaked seeds were then placed into moistened seedling sponges and incubated at room temperature for a duration of 3 days. After this initial incubation period, the seedlings were transferred to containers for an additional 3-day incubation period, ensuring the intact growth of their roots. Finally, the viable seedlings were transplanted into transparent plastic containers for cultivation, with seven plants per pot. Each container was filled with diluted 900 mL of general-purpose hydroponic nutrient solution (general-purpose hydroponic nutrient solution: distilled water = 1:300). This experiment was set up with two test groups (heavy metal chromium + 1% inoculated strain treatment) and control group (adding heavy metal chromium + no inoculated strain). The results of the pre-test for normal growth of radish seedlings showed that radish seedlings could not grow under high concentration of Cr(VI) (30 mg·L−1) (control and experimental groups). Therefore, a range of Cr(VI) concentrations in the growing environment of radish seedlings was screened. Radish seedlings were treated with 0, 1, 2, 3, 4, 5, 6, and 7 mg·L−1 Cr(VI). Radish seedlings were placed in an artificial culture laboratory with a day/night time of 16/8 h, a day/night temperature of 26 ± 2.5 °C/18 ± 2 °C and a light intensity of 25,000 lux. During the cultivation of radish seedlings, nutrient solution was added every 3 days to ensure adequate nutrition. After 20 days of incubation, the radish seedlings were collected, and the plant height and fresh weight of the radish leaves were measured after draining the water from the surface.

Chlorophyll and biochemical analyses

Determination of chlorophyll levels

Radish seedlings were rinsed with sterile water, and chlorophyll content was measured using the ethanol-acetone method. Chlorophyll was extracted by adding 0.1 g of clipped leaves to the mixed extraction solution (acetone: anhydrous ethanol = 1:1) (Zhang, 1986). The leaves were sealed and placed in the dark in an incubator at a constant temperature of 30 °C. The leaves were observed until they turned completely white. Optical density (OD) values of the samples were measured at 663 nm (chlorophyll a) and 645 nm (chlorophyll b), with the mixed extraction solution used as the blank. The chlorophyll a and chlorophyll b concentrations obtained, along with their corresponding OD values, are listed below:

(3) D663=82.04Ca+9.27Cb

(4) D645=16.75Ca+45.60Cb

D663 and D645 are the optical densities at 663 and 645 nm, respectively; Ca and Cb are the concentrations of chlorophyll a and chlorophyll b in g·L−1, respectively. This can be obtained after solving the equation:

(5) Ca=12.7D663−2.59D645

(6) Cb=22.9D645−4.67D663

(7) Ca+b=20.3D645+8.04D663

where Ca and Cb are the concentrations of chlorophyll a and chlorophyll b, respectively, and Ca+b is the total concentration of chlorophyll a and chlorophyll b in mg·L−1. The unit weight of the measured leaves can be further calculated according to the following formula:

(8) chlorophyllcontent(mg⋅L−1)=C∗VA∗1,000

where C is the chlorophyll concentration (mg·L−1), V is the total volume of extract (mL), and A is the fresh weight of the leaf (g).

Determination of total protein, malondialdehyde, superoxide dismutase and catalase

The leaves were washed 2–3 times with 0.1 mol·L−1 phosphate buffer (pH = 7.4) at 4 °C. Next, the leaf surface were blotted dry with filter paper and then cut into pieces. Subsequently, 0.1 g of leaf pieces were placed in a frozen 5 mL homogenizing tube, and homogenizing medium (0.1 mol·L−1 phosphate buffer (pH 7.4) at 4 °C) was added in a weight (g) to volume (mL) ratio of 1:4. The mixture was ground in an ice bath to make a 20% homogenate. Subsequently, this homogenate was transferred to a high-speed freezing centrifuge (3,500 r·min−1) and centrifuged for 10 min. Total protein (TP) content was determined using Coomassie blue staining with albumin as a standard (Bradford, 1976), malondialdehyde (MDA) content was determined by the thiobarbituric acid (0.1%) method (TBA) (Heath & Packer, 1968), and superoxide dismutation (SOD) and catalase (CAT) contents were determined by the hydroxylamine and ammonium molybdate methods, respectively (Beers & Sizer, 1952; Beauchamp & Fridovich, 1971).

Determination of chromium content in radish seedlings and hydroponic solutions

Radish seedlings were rinsed, their surfaces were wiped dry, and leaves and taproots were dried in an oven at 70 °C. Next, 0.1 g of leaves and 0.1 g of taproots were weighed and transferred to a crucible for elimination. Total chromium content in leaves and taproots were determined using an inductively coupled plasma spectrometer (ICPS) (Faisal & Hasnain, 2005). At the end of the incubation (20 days in vitro), the hydroponic solution was taken from the container and filtered through a 0.45 μm filter membrane, and the residual Cr(VI) ions were measured using the diphenyl carbamide method. One milliliter of the hydroponic solution was aspirated into a 50 mL crucible for digestion, and the total chromium content was determined using ICPS. The digestion of leaves, taproots, and hydroponic solutions was carried out using the electrothermal plate digestion method (HCl-HNO3 system) as previously outlined.

Statistical analysis

Data were represented as mean values and standard error. All data were obtained from three replicate experiments. Data were processed using SPSS 26 and analyzed by ANOVA (P < 0.05). Data visualization was performed using Origin 2019b. FITR and XPS results were analyzed using Omnic and Avantage software, respectively.

Results

Identification and developmental tree analysis of the LBA36 strain

The colony morphology of the LBA36 strain is shown in Fig. 2A, exhibiting a colony size of 2–4 mm. The colonies possessed neat edges and a uniform surface, and appeared white, non-sticky, and without odor. Under an optical microscope (Fig. 2B) the cells appear as a rod-shaped, gram-positive bacterium. SEM analysis revealed that LBA36 strain was long, rod-shaped bacterium measuring approximately 0.5 × 1.0 μm in size (Fig. 2C).

Figure 2 Micromorphology of the LBA36 strain and phylogenetic tree.

(A) Colony morphology of the LBA36 strain on a LB agar plate. (B) Gram staining of LBA36 under an optical microscope. (C) Scanning electron microscopy (SEM) analysis of the LBA36 strain. (D) Phylogenetic tree of LBA36. (The 16S rDNA gene amplification sequence of LBA36 is shown in the Data S1).

The phylogenetic tree of LBA36 strain is shown in Fig. 2D, and the degree of similarity between LBA36 strain and Bacillus toyonensis BCT-7112T (CP006863) was 99.5% (Data S1). 16S rDNA sequence homology revealed that LBA36 strain belongs to the genus Bacillus.

Chromium reduction capacity, adsorption efficiency, and growth curves of LBA36

The growth of LBA36 varied under different Cr(VI) concentrations. As shown in Fig. 3A, the logarithmic growth period of the LBA36 occurred between 3–18 h in the absence of Cr(VI) stress. The onset of the logarithmic growth phase of LBA36 was delayed with increasing Cr(VI) concentrations. The logarithmic growth phase of LBA36 occurred between 18–57 h when grown in media containing 60 mg·L−1 Cr(VI). At 90 mg·L−1 of Cr(VI), the growth of LBA36 strain was markedly inhibited. The MIC of Cr(VI) was 90 mg·L−1.

Figure 3 Growth curves of LBA36 and Cr(VI) removal under different conditions.

(A) Growth curve of the LBA36 strain. (B) Cr(VI) reduction and adsorption efficiency of the LBA36 strain. (C) Effect of time on the Cr(VI) treatment efficiency of the LBA36 strain. (D) Reducing the capacity of different fractions of the LBA36 strain. (EG: the experimental group (Cr(VI)-containing medium inoculated with the LBA36 strain); CG: the control group (Cr(VI)-containing medium without LBA36 strain inoculation)). Data are presented as the mean + standard error of three replicates (P < 0.05).

The reduction and adsorption efficiencies of LBA36 decreased with an increase in the Cr(VI) concentration. The maximum reduction rate of LBA36 was greater than 90%, and the total Cr adsorption rate was about 9% (Fig. 3B). As shown in Fig. 3C, under stress of 30 mg·L−1 Cr(VI), the reduction rate of Cr(VI) increased with time until it reached a certain limit, beyond which it stabilized at approximately 97%. This can be attributed to the inherent reduction characteristics of the strain itself. The Cr adsorption efficiency of the strain initially increased and then decreased with time, and the maximum adsorption rate was 8.8%, observed at 32 h. This may be because LBA36 reached its maximum Cr adsorption capacity, leading to a change in the surface or internal group of the strain. This may have triggered a change in the adsorption capacity, potentially resulting in desorption.

The Cr(VI) reduction ability of different components of the strain was as follows: extracellular material > resting cell > permeabilized cell > intracellular material, indicating that the reduction of Cr(VI) mainly took place in the extracellular area. The reduction rate of the extracellular material was 31.46% (Fig. 3D). The reduction capacity of permeabilized cells for Cr(VI) was lower than that of resting cells, which indicated that the cell membrane played an important role in the reduction process. The permeability of the cell membrane treated with toluene and Triton X-100 was noticeably increased, and Cr(VI) could penetrate the cell more easily, which altered the cells’ ability to reduce Cr(VI).

FTIR and XPS studies

The infrared spectra of LBA36 after 36 h of incubation at Cr(VI) concentrations of 0, 30, and 60 mg·L−1 are shown in Fig. 4A. Owing to the different concentrations of Cr(VI), the FTIR spectra of LBA36 exhibited marked changes. The strong broad peak at 3,301.79 cm−1 is the superposition of O-H and-NH telescopic vibration peaks; the C-H telescopic vibration peak of aliphatic CH2 can be seen at 2,934.05 cm−1; the C=O telescopic vibration peak of ester can be seen at 1,736.68 cm-1; the C=O telescopic vibration peak of amide I band is observed at 1,655.02 cm−1; the C-H asymmetric bending vibration peak of CH3 group can be seen at 1,450.92 cm−1; and the C-O stretching vibration peak of carboxylic acid can be seen at 1,389.41 cm−1. These absorption peaks were shifted to high wave numbers, indicating that the numerous bacterial cell surface functional groups, including surface methyl-, amide I-and amide II groups may be involved in the reduction and adsorption of Cr(VI), preventing the entry of Cr(VI) into the cells, reducing the toxic effect of Cr(VI), and improving the tolerance of LBA36 to Cr(VI). The bending vibration peak of amide II with N-H at 1,546.21 cm−1, the telescopic vibration peak of C-N at 1,242.14 cm−1, the superposition of the C-O peak of ester and the C=S peak of sulfur carbonyl at 1,060.18 cm−1, and the absorption peak of PO43− at 628.66 cm−1 were all shifted to lower wave numbers, indicating that they may have altered the toxicity of Cr(VI) to LBA36. These shifting of peaks to lower wave numbers indicates that Cr(VI) may have changed the chemical properties and concentration levels of carbohydrates, esters, proteins, and other biomolecules in LBA36. In LBA36, OH, CH, C-N, C=O, NH, C=S are usually part of proteins and sugars.

Figure 4 Characterization of the LBA36 strain before and after Cr(VI) stress.

(A) FITR spectra of the LBA36 strain at different Cr concentrations (* indicates blue shift of peaks, # indicates red shift of peaks). (B–D) XPS profiles of the LBA36 strain before and after exposure to 30 mg·L −1 Cr(VI). (FTIR, Fourier-transform infrared spectroscopy; XPS, X-ray photoelectron spectroscopy).

The valence states of Cr on the cell surface were analyzed by XPS (Figs. 4B, 4C). Compared with the LBA36 treated with 0 mg·L−1 Cr(VI), the strain treated with 30 mg·L−1 Cr(VI) exhibited relatively lower C-(O/N) content, relatively higher C-O, C=O and O-C-O content. The relative content of C=O in polysaccharides increased slightly and the relative content of C-O increased considerably. Combined with the infrared spectra, the characteristic peaks of polysaccharides and proteins in the LBA36 cells drifted when treated with 30 mg·L−1 Cr(VI), indicating that the functional groups in polysaccharides and proteins interacted with Cr(VI).

As shown in Fig. 4D, the peaks appeared at 586.5–588 eV (Cr2p1/2) and 576.88–578.76 eV (Cr2p3/2), and LBA36 reduced Cr(VI) to Cr(III). The valence states of Cr present on the surface of LBA36 were +6 and +3, and the ratio of Cr(VI) to Cr(III) was 1:2. Thus, the ratio of adsorbed Cr(VI) to Cr(III) on the surface of LBA36 cells was 1:2.

Determination of fresh weight, dry weight, and length of radish seedlings

The height and weight of plant seedlings are important indicators of plant growth and development status. The degree of toxicity of Cr(VI) to radish seedlings was determined by measuring the germination rate, plant height, fresh weight, and root length of radish seedlings under Cr(VI) stress. As shown in Fig. 5A, in the control group (seeds + sterile distilled water), with increasing concentrations of Cr(VI), the germination rate of seeds decreased, reaching around 50%; in the experimental group (seeds + LBA36 bacterial solution), the concentration of Cr(VI) also showed a decreasing trend, but the overall germination rate of the experimental group was higher than that of the control group. The germination rate of the experimental group was higher than that of the control group. The germination rate of the experimental group increased by 17.74%, 16.39%, 13.33%, 13.56%, 15.52%, 9.4% and 10% at different Cr(VI) concentrations, respectively. Figure 5B shows that the weight of radish seedlings in both the control and the experimental groups initially increased and then decreased with increasing Cr(VI) concentrations. Figure 5C revealed that inhibition of root length of radish seedlings increased with increasing Cr(VI) concentrations in both control and experimental groups. Root length of radish seedlings in experimental groups eventually became larger than that of the control group due to the presence of LBA36. Figure 5D shows that with an increase in Cr(VI) concentration, the stem length of radish seedlings in the control group and experimental group initially increased and then decreased. When the Cr(VI) concentration was greater than 100 mg·L−1, the stem length of radish seedlings in the experimental group was higher than that in the control group, and the inhibitory effect became considerably significant.

Figure 5 Growth of radish seedlings after in vivo and in vitro experiments on fresh weight, dry weight and length.

In vivo experiment: (A) Radish seedling seed germination. (B) Weight of radish seedlings. (C) Radish seedling root length. (D) Radish seedling diameter. In vitro experiment: (E) Weight of radish seedlings. (F) Radish seedling root length. (G) Effect of inoculation with LBA36 strain on the growth condition of radish seedlings. (EG: the experimental group (Inoculated with the LBA36 strain); CG: the control group (Inoculated without the LBA36 strain). Data are presented as the mean + standard error of three replicates. Different letters within a column indicate significant difference between the treatments (P < 0.05).

As shown in Figs. 5E and 5F, radish seedling growth and development decreased with an increase in Cr(VI) concentration. The fresh weight of the control group gradually decreased, and plant height initially increased and then decreased, indicating that small amounts of Cr(VI) or Cr(III) in the environment promote the growth of radish seedlings. In the experimental group, radish seedlings grown in 3 and 4 mg·L−1 Cr(VI) exhibited the best growth, and the highest fresh weight and plant height values.

Figure 5G shows that the control and experimental groups grew well, and that the seedlings in the experimental group inoculated with LBA36 strain were taller, with wider leaves and longer taproots. In the control group, the size of radish seedlings, rootstocks, and leaves decreased with increasing Cr(VI) concentrations. Furthermore, the leaves became yellower with increasing Cr(VI) concentrations. In the experimental group, with increasing Cr(VI) concentrations, the taproots of radish seedlings gradually thickened, the diameter of the leaves increased, and the rootstock became shorter and smaller. This phenomenon indicated that LBA36 could promote the growth and development of radish seedlings, but after Cr(VI) exceeded 3 mg·L−1, LBA36 was unable to prevent Cr(VI) from poisoning radish seedlings, and they suffered from stunted growth and development, yellowing of leaves, and impaired photosynthesis.

Under Cr(VI) stress, the presence of LBA36 could enhance fresh weight and promote rootstock growth. Compared with CG0-CG7, the fresh weight of EG0-EG7 radish seedlings increased by 16.75%, 66.81%, 73.29%, 85.17%, 87.73%, 81.4%, 85.47% and 87.81%, respectively; and the height of the plants increased by 30.45%, 35.96%, 32.65%, 37.5%, 45.47%, 32.58%, 28.09% and 37.07% respectively. Radish seedlings in the experimental group had longer taproots, more leaves, and wider and greener leaves.

Effect of LBA36 strain on chlorophyll and biochemical properties in radish seedlings

Figure 6A shows that with increasing Cr(VI) concentration, the chlorophyll content of leaves in the control group and experimental group decreased significantly. The effect of LBA36 on TP content in leaves of radish seedlings under Cr(VI) stress is shown in Fig 6B. As Cr(VI) levels increased, its inhibitory effect on the TP content in leaves became increasingly apparent. At the same Cr(VI) concentration, the TP content was higher in the experimental group. As shown in Fig. 6C, the MDA content in the leaves increased significantly with Cr(VI) concentration. The leaves in the control group gradually turned yellowed, indicating an increase in cell membrane lipidation in the leaves of radish seedlings, and the highest concentration of Cr(VI) caused serious damage to the cell membrane of the leaves. Figure 6D shows the effect of LBA36 on SOD enzyme activity in leaves of radish seedlings under Cr(VI) stress. As Cr(VI) concentration increased, SOD enzyme activity increased and then decreased. Lower concentrations of Cr(VI) enhanced SOD enzyme activity, and high concentration of Cr(VI) significantly inhibited SOD activity. Figure 6E shows the effect of LBA36 on CAT enzyme activity in the leaves of radish seedlings under Cr(VI) stress. As Cr(VI) concentrations increased, CAT activity was significantly inhibited in the leaves of radish seedlings. At similar Cr(VI) concentrations, CAT content in the experimental group markedly differed from that of the control group across the concentration gradient range of 1–6 mg·L−1. In the control group, the CAT content decreased as Cr(VI) concentration increased, whereas, in the experimental group, CAT concentration increased with increasing Cr (VI) levels.

Figure 6 Effect of the LBA36 strain on the content of chlorophyll, TP, MDA, SOD and CAT.

(A) Effect of LBA36 strain inoculation on chlorophyll content in leaves of radish seedlings. (B) Effect of LBA36 strain inoculation on TP content in the leaves of radish seedlings. (C) Effect of LBA36 strain inoculation on MDA content in leaves of radish seedlings. (D) Effect of LBA36 strain inoculation on SOD content in the leaves of radish seedlings. (E) Effect of LBA36 strain inoculation on CAT content in the leaves of radish seedlings. (TP, Total Proteins; MDA, Malondialdehyde; SOD, Superoxide dismutation; CAT, Catalase; EG, the experimental group (Inoculated with the LBA36 strain); CG, control group (no LBA36 strain inoculation). Data is presented as mean + standard error of three replicates. Different letters within a column indicate significant difference between the treatments (P < 0.05).

Changes in Cr(VI) content in radish seedlings in vivo and in hydroponic solution

As shown in Fig. 7A, under similar Cr(VI) concentrations, Cr accumulation in taproots was higher in the experimental group than that in the control group. In contrast, Cr accumulation was higher in leaves in the control group than that in the experimental group (Fig. 7B). This may be due to the fact that LBA36 reduces environmental Cr(VI) to Cr(III), which decreases Cr uptake and accumulation in the leaves of radish seedlings. Furthermore, at the same Cr(VI) concentration, Cr accumulation capacity varied in different parts of radish seedlings. Cr content in the roots of radish seedlings was much higher than that in their leaves, indicating that Cr primarily accumulated in the roots of radish seedlings and exhibited limited transference to the leaves. A related study demonstrated that certain plants, such as water hyacinth and lobelia, accumulated more Cr in their roots than their stems and leaves (Batool, Tabassum & Ali, 2017).

Figure 7 Changes in Cr(VI) content in radish seedlings and hydroponic solutions after Cr(VI) stress.

(A) Cr content in taproots of radish seedlings inoculated with LBA36. (B) Cr content in leaves of radish seedlings inoculated with LBA36. (C) Effect of LBA36 strain inoculation on Cr(VI) and total Cr content in culture broth during radish seedling cultivation. (EG: the experimental group (inoculated with LBA36); CG: the control group (no LBA36 inoculation). Data are represented as mean + standard error of three replicates. Different letters within a column indicate significant difference between the treatments (P < 0.05).

Cr(VI) was completely reduced in the control group within the concentration range of 1–2 mg·L−1, possibly attributed to the presence of reducing substances in the hydroponic solution or the release of such substances through the plant’s own protective mechanisms (Fig. 7C). At a concentration of 1–5 mg·L−1, Cr(VI) was completely reduced in the experimental group, and at 6- and 7 mg·L−1, the Cr(VI) concentration was only 0.01 and 0.08 mg·L−1, respectively. At the same Cr concentration, the total Cr adsorption rate of the experimental groups exhibited varying degrees of increase. Overall, LBA36 reduced a large portion of Cr(VI) in the environment and promoted the uptake of Cr ions from aqueous solutions by the roots of radish seedlings.

Discussion

In this study, we investigated the Cr reduction performance of Cr-tolerant bacterium LBA36 and the protective properties of the strain on plants. The Cr reduction and adsorption efficacies of the strain was investigated and Cr reduction by different components of the strain was assessed. In addition, growth of radish seedlings, biochemical reactions, and Cr accumulation in radish seedlings were studied.

Cr-tolerant microbes are scarce in the natural environment, and as such, Cr-tolerant microbial strains need to be screened from polluted sources. In the past, Cr-tolerant strains were isolated from polluted sources and their potential in remediation of soil Cr(VI) pollution was explored. Upadhyay et al. (2017) examined and characterized bacteria with the high potential to reduce Cr(VI) from contaminated soils, and investigated the effects of different Cr treatments on the growth and morphology of the bacteria to enhance bioremediation efforts.

The prerequisite for the joint remediation of Cr(VI) pollution by microorganisms and plants relies on their ability to promote growth and mutually benefit from each other’s presence (Guo et al., 2020). In this study, LBA36 from a combined microbial-plant system was inoculated into a hydroponic solution. LBA36 enhanced the resistance of radish seedlings to Cr(VI), reduced Cr(VI) toxicity, and markedly promoted radish seedling growth (Gupta et al., 2020). In a similar study by Peng et al. (2021), two strains of Cr-reducing bacteria with high Cr tolerance were identified which promoted the growth and Cr uptake of tall fescue, thereby enhancing their phytoremediation efficiency. A study by Shi et al. (2023) reported Cr(VI) resistance of AN-B15 and its ability to reduce Cr(VI). Furthermore, the group hydroponically cultivated wheat seedlings and found that inoculation with AN-B15 increased root length and stem length by 11.1% and 5.7%, respectively, and their fresh weight by 55.5% and 18.8%, respectively.

Our study also reported similar results. We demonstrated that LBA36 inoculation increased the fresh weight of radish seedlings increased by 87.73% and plant height by 45.57%. A study by Sun (2019) investigated the stress response and Cr accumulation in wheat, radish, cucumber, cabbage, rapeseed and lettuce. This study offers theoretical underpinning for the safe cultivation of crops in low to moderate Cr contamination.

The present study found radish seedlings under Cr(VI) stress exhibited an optimal Cr concentration, as indicated by the fact that certain physiological parameters, such as plant height and fresh weight of radish seedlings, were maximized at this concentration. However, as Cr(VI) concentration increased, these physiological parameters decreased. This may be attributed to the fact that, at the optimum Cr(VI) concentration, non-protein nitrogen content in the radish seedlings was reduced and the as Cr levels increased, protein content decreased (Tiwari, Singh & Rai, 2013).

Cr-induced reduction in plant developmental growth and yield is mainly due to imbalance in nutrient uptake and translocation, inefficient selective inorganic nutrient uptake, and oxidative damage to sensitive tissues in plants (Chen et al., 2016; Maqbool et al., 2018). Increased Cr levels in plants leads to changes in ultrastructure (Sallah-Ud-Din et al., 2017), and increased oxidative stress, electrolyte leakage and MDA concentrations, while altering activities of antioxidant enzyme, such as SOD, CAT, peroxidase (POD), and ascorbate peroxidase (APX) (Zaheer et al., 2020).

Cr(VI) degrades photosynthetic pigments, reduces chlorophyll synthesis, inactivates Calvin cycle enzymes, reduces CO2 fixation, and impairs the photosynthetic electron transport system in plants (Shanker et al., 2005). The higher chlorophyll content in the experimental group at the same Cr(VI) concentration may be attributed to reduction of Cr(VI) to Cr(III) by LBA36, which reduced toxicity of Cr(VI) to radish seedlings, resulting in an increase in chlorophyll content. The lower MDA content in the experimental group at the same Cr(VI) concentration may be attributed to the reduction of Cr(VI) in the environment by LBA36, which decreased lipid peroxidation of the cell membranes in the radish. SOD and POD are important enzymes in crops’ antioxidant defense mechanism. SOD is the most critical enzyme in plants’ resistance against oxidative stress, acting as the first line of defense and reducing the toxic effects induced by elevated ROS levels. In general, low Cr(VI) concentrations enhance antioxidant enzyme activity, while high concentrations of Cr(VI) inhibits antioxidant enzyme activity (Sun, 2019). The CAT content in the experimental group increased with increase in Cr(VI) concentration, possibly because LBA36 reduced the Cr(VI) in the environment without damaging the antioxidant defense system of radish seedlings, thereby alleviating Cr(VI)-induced oxidative stress. Analysis of chlorophyll, TP, MDA, SOD and CAT contents of radish seedlings revealed that Cr(VI) decreased the levels of chlorophyll and various proteins, causing cellular damage to radish seedlings. However, the phytotoxicity of Cr(VI) to radish seedlings could be partially mitigated by inoculation with LBA36. Wani & Khan (2010) isolated Bacillus sphaericus PSB10, and demonstrated that it significantly improved the growth, nodulation, chlorophyll level, seed yield and grain protein levels of chickpea crop grown in the presence of varying Cr concentration as compared to plants grown in the absence of bioinoculants. Ali et al. (2022), in their study on microbial-citric acid assisted phytoremediation of Cr in castor plants, analyzed the influence of citric acid and Staphylococcus aureus on the remediation of Cr(VI) contaminated soils and reported a significant decrease in chlorophyll content of the plant with the increase in Cr levels, which is in agreement with our findings. Solá et al. (2021) assessed the effect of Streptomyces Waksman & Henrici and Zea mays L. system on the dissipation of Cr(VI) and lindane from co-contaminated soils, and found that plants inoculated with Streptomyces z38 exhibited decreased MDA concentrations when grown in Cr(VI), suggesting that the bacteria can activate plant defense mechanisms.

In our study, we observed that Cr accumulates mainly in the roots of radish seedlings and was not readily translocated to the leaves. This finding corroborated those of Gomes et al. (2017), who measured Cr(VI) in the roots, stems, leaves, and seeds of wheat plants and showed that the element moves from the bottom to the top. The accumulation of Cr in plant roots may result from the formation of insoluble compounds within the plant. Additionally, it has been reported that the increased accumulation of Cr(VI) in roots may be attributed to its enhanced vesicular segregation in root cells, which serves to limit the toxic potential of Cr (Singh et al., 2013).

Conclusion

In this study, a Cr(VI)-resistant bacterial strain (LBA36) was isolated from heavy metal contaminated soil in the mining area, and identified as Bacillus toyonensis with a MIC of 90 mg·L−1. Our study revealed that Cr(VI) reduction predominantly took place extracellularly, and the reduction and adsorption efficiencies were 97.95% and 8.8%, respectively. The analysis of various characteristics showed that various functional groups, including-OH in LBA36, CH in aliphatic CH2, C=O in aliphatic chain, C=O in amide I band, C=O in amide II band, NH in amide II band, C-N in amide III band, C-H in CH3 group in amide III band, C-O in carboxylic COOH in amide III band, C-N in amide III band, C-O in esters, and C=S in sulfur carbonyls, and PO43− maybe involved in the removal of Cr(VI). Cr was present in its +6 and +3 valence states on the surface of the bacteria. Hydroponic experiments on radish seedlings revealed that Cr(VI) toxicity increased with Cr(VI) concentration in radish seedlings. The LBA36 strain was able to reduce Cr(VI) in the environment, increase the uptake of Cr by radish seedlings’ taproots, and reduce the uptake and utilization of chromium by leaves, potentially protecting the radish seedlings. The LBA36 strain holds promise for the remediation of Cr(VI)-contaminated water and soil. Future investigations must explore aspects such as the adhesion of bacterial cells to plants and plant secretions that enhance bacterial cell growth.

Supplemental Information

Supplemental Information 1 16S rDNA gene sequencing results of experimental strains.

Supplemental Information 2 Original data.

We are grateful to the College of Chemistry and Chemical Engineering of Henan University of Science and Technology for the use of the equipment or facilities provided.

Additional Information and Declarations

Competing Interests

Author Contributions

DNA Deposition

Data Availability

The authors declare that they have no competing interests.

Aobo Tan conceived and designed the experiments, performed the experiments, analyzed the data, prepared figures and/or tables, authored or reviewed drafts of the article, and approved the final draft.

Hui Wang conceived and designed the experiments, analyzed the data, prepared figures and/or tables, authored or reviewed drafts of the article, and approved the final draft.

Hehe Zhang analyzed the data, authored or reviewed drafts of the article, and approved the final draft.

Longfei Zhang performed the experiments, authored or reviewed drafts of the article, and approved the final draft.

Hanyue Yao performed the experiments, prepared figures and/or tables, and approved the final draft.

Zhi Chen conceived and designed the experiments, authored or reviewed drafts of the article, and approved the final draft.

The following information was supplied regarding the deposition of DNA sequences:

The sequences described are available in the Supplemental File.

The following information was supplied regarding data availability:

The raw data are available in the Supplemental File.

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
