# Peer review of "Reduction of Cr(VI) by Bacillus toyonensis LBA36 and its effect on radish seedlings under Cr(VI) stress"

_PeerJ, doi:10.7717/peerj.18001_

## Round 0.1 · original submission · Major Revisions

Please revise the manuscript based on the reviewers' comments.

Reviewer 1 ·

Basic reporting

Dear editor,
This is a well-written paper with interesting results focus on the potential of microbial and phytoremediation in the treatment of heavy metal toxicity. For the benefits of the reader, however, few points need to make modification. They are as follows:
1.The author needs to focus on making his English more readable. For example, in line 131 “To investigate the heavy metal removal efficacy of the LBA36 strain at different Cr(VI) concentrations and the effect of time on Cr(VI) removal by LBA36 strain, the following experiments were performed: (1) LBA36 strain was cultured in medium containing 30, 60, and 90 mg·L-1 of Cr(VI). Cr(VI)-containing medium without the strain inoculation was used as a negative control. Three parallel groups were set up for each concentration. Samples were collected and centrifuged after 32 h of incubation (8000 r·min-1, 10min), and 1 mL of supernatant was aspirated for digestion. (2) The LBA36 strain was cultivated in a medium containing 30 mg·L-1 Cr(VI).”
2. In Abstract section, the key data should be laid out.
3. In Introduction section, the authors didn’t mention the research material radish and why chose radish in hydroponic experiments.
4. In Material and method section line 133, he author should explain why choose the medium containing 30, 60, and 134 90 mg·L-1 of Cr(VI).
5. In Radish seedling hydroponic experiment, which type of radish did the author choose, hybrid, inbreed or cultivar? The name of the radish? The detail of the material shoud be offered. Why did the author only used one radish?
6. In line 197, Why did the author choose Cr(VI) solution at graded concentrations (0, 30, 60, 90, 100, 200, and 400 mg·L-1)? and what’s the purpose of incubating seed in the dark at 30℃ for 7 days.
7. In line 214, explain why choose seeds treated with 0, 1, 2, 3, 4, 5, 6, and 7 mg·L-1 Cr(VI).
8. In line 215, the details of the cultivation of radish seedlings conditions should be gave, such as temperature, light intensity.
9. In line 242, Determination of TP, MDA, SOD, CAT, use the full when frist mentioned.
10. In line 256, there is no rhizomes in radish, we used taproot in radish.
11. In Statistical analysis, the author said “data were represented as mean values and standard error”. But in Fig. 3, 4, 5 and 6, it is “Data arepresented as the mean + standard deviation of three replicates”. We used standard error when the sample is less than 30(n<30),and used standard deviation when the sample is more than30(n>30).
12. In Fig. 4, the font size is inconsistent.
It clear that the authors need to do a modification work on the manuscript before it would be suitable.

Experimental design

no comment

Validity of the findings

This is a well-written paper with interesting results focus on the potential of microbial and phytoremediation in the treatment of heavy metal toxicity.

·

Basic reporting

The manuscript is well-written, but missing some references.

Experimental design

The number of replicates was not stated

Validity of the findings

The conclusions are well stated.

Additional comments

Some references for the methods used are missing.

Reviewer 3 ·

Basic reporting

In this manuscript, the authors isolated chromium-tolerant soil bacteria with the ability to reduce Cr(VI) to Cr(III). Standards were followed in terms of experimental design. Background information and references are well presented and cited. Overall, the manuscript is highly interesting. I have only a few specific comments. There are a few places in the manuscript where the rules of abbreviation are not followed. The references need to be improved according to the requirements of the journal.

Experimental design

no comment

Validity of the findings

no comment

Additional comments

1.Line 163: The descriptions of the cell components do not correspond to the y-axis plot captions in Figure 3D.
2.Line 171: “20 s” “30 s”.
3.Lines 210-211: Please describe the specific components of the universal hydroponic solution.
4.Line 276: Replace “um” with “μm”.
5.Line 180: In the section “Chemical analysis and characterization”, why did the authors set the Cr concentrations at 30 and 60 mg·L-1?
6.Line 461: “CO2”
7.Write the full term before writing the abbreviation. After that, it is not necessary to write the full term in the same section (eg. line 34: Chromium)
8.Lines 91, 94, 97: Reference formatting needs to be revised.

---

## Round 0.2 · accepted · Accept

After reviewing the point-by-point responses, I believe the authors have satisfactorily addressed all the reviewers' comments, and the manuscript can be accepted in its current form.

Reviewer 3 ·

Basic reporting

no comment

Experimental design

no comment

Validity of the findings

no comment

Additional comments

The authors made changes to the manuscript based on the reviewers' comments. I have no further comments on the revised manuscript.